# EMTL: A GENERATIVE DOMAIN ADAPTATION APPROACH

## ABSTRACT

We propose an unsupervised domain adaptation approach based on generative models. We show that when the source probability density function can be learned, one-step Expectation–Maximization iteration plus an additional marginal density function constraint will produce a proper mediator probability density function to bridge the gap between the source and target domains. The breakthrough is based on modern generative models (autoregressive mixture density nets) that are competitive to discriminative models on moderate-dimensional classification problems. By decoupling the source density estimation from the adaption steps, we can design a domain adaptation approach where the source data is locked away after being processed only once, opening the door to transfer when data security or privacy concerns impede the use of traditional domain adaptation. We demonstrate that our approach can achieve state-of-the-art performance on synthetic and real data sets, without accessing the source data at the adaptation phase.

## 1 INTRODUCTION

In the classical supervised learning paradigm, we assume that the training and test data come from the same distribution. In practice, this assumption often does not hold. When the pipeline includes massive data labeling, models are routinely retrained after each data collecion campaign. However, data labeling costs often make retraining impractical. Without labeled data, it is still possible to train the model by using a training set which is relevant but not identically distributed to the test set. Due to the distribution shift between the training and test sets, the performance usually cannot be guaranteed.

Domain adaptation (DA) is a machine learning subdomain that aims at learning a model from biased training data. It explores the relationship between source (labeled training data) and target (test data) domains to find the mapping function and fix the bias, so that the model learned on the source data can be applied in target domain. Usually some target data is needed during the training phase to calibrate the model. In *unsupervised* domain adaptation (UDA) only *unlabeled* target data is needed during training phase. UDA is an appealing learning paradigm since obtaining unlabeled data is usually easy in a lot of applications. UDA allows the model to be deployed in various target domains with different shifts using a single labeled source data set.

Due to these appealing operational features, UDA has became a prominent research field with various approaches. Kouw & Loog (2019) and Zhuang et al. (2020) surveyed the latest progress on UDA and found that most of the approaches are based on discriminative models, either by reweighting the source instances to approximate the target distribution or learning a feature mapping function to reduce the statistical distance between the source and target domains. After calibrating, a discriminative model is trained on the adjusted source data and used in target domain. In this workflow, the adaptation algorithm usually have to *access the source and target data simultaneously*. However, accessing the source data during the adaptation phase is not possible when the source data is sensitive (for example because of security or privacy issues). In particular, in our application workflow an industrial company is selling devices to various service companies which cannot share their customer data with each other. The industrial company may contract with one of the service companies to access their data during an R&D phase, but this data will not be available when the industrial company sells the device (and the predictive model) to other service companies.

In this paper we propose EMTL, a *generative* UDA algorithm for binary classification that does not have to access the source data during the adaptation phase. We use density estimation to estimate the joint source probability function $p^{\mathrm{s}}(\mathbf{x}, y)$ and the marginal target probability function $p^{\mathrm{t}}(\mathbf{x})$ and use them for domain adaption. To solve the data security issue, EMTL decouples source density estimation from the adaptation steps. In this way, after the source preprocessing we can put away or delete the source data. Our approach is motivated by the theory on domain adaptation (Ben-David et al., 2010) which claims that the error of a hypothesis $h$ on the target domain can be bounded by three items: the error on the source domain, the distance between source and target distributions, and the expected difference in labeling functions. This theorem motivated us to define a mediator density function $p^{\mathrm{m}}(\mathbf{x}, y)$ i) whose conditional probability $y|\mathbf{x}$ is equal to the conditional probability of the source and ii) whose marginal density on $\mathbf{x}$ is equal to the marginal density of the target. We can then construct a Bayes optimal classifier on the target domain under the assumption of *covariate shift* (the distribution $y|\mathbf{x}$ is the same in the source and target domains).

Our approach became practical with the recent advances in (autoregressive) neural density estimation (Uria et al., 2013). We learn $p^{\mathrm{m}}(\mathbf{x}, y)$ from $p^{\mathrm{s}}(\mathbf{x}, y)$ and $p^{\mathrm{t}}(\mathbf{x})$ to bridge the gap between the source and target domains. We regard the label on the target data as a latent variable and show that if $p^{\mathrm{s}}(\mathbf{x}|y=i)$ be learned perfectly for $i \in \{0, 1\}$, then a one-step Expectation–Maximization (and this is why our algorithm named EMTL) iteration will produce a density function $p^{\mathrm{m}}(\mathbf{x}, y)$ with the following properties on the target data: i) minimizing the Kullback–Leibler divergence between $p^{\mathrm{m}}(y_i|\mathbf{x}_i)$ and $p^{\mathrm{s}}(y_i|\mathbf{x}_i)$; ii) maximizing the log-likelihood $\sum \log p^{\mathrm{m}}(\mathbf{x}_i)$. Then, by adding an additional marginal constraint on $p^{\mathrm{m}}(\mathbf{x}_i)$ to make it close to $p^{\mathrm{t}}(\mathbf{x}_i)$ on the target data explicitly, we obtain the final objective function for EMTL. Although this analysis assumes a simple covariate shift , we will experimentally show that EMTL can go beyond this assumption and work well in other distribution shifts.

We conduct experiments on synthetic and real data to demonstrate the effectiveness of EMTL. First, we construct a simple two-dimensional data set to visualize the performance of EMTL. Second, we use UCI benchmark data sets and the Amazon reviews data set to show that EMTL is competitive with state-of-the-art UDA algorithms, without accessing the source data at the adaptation phase. To our best knowledge, EMTL is the first work using density estimation for unsupervised domain adaptation. Unlike other existing generative approaches (Kingma et al., 2014; Karbalayghareh et al., 2018; Sankaranarayanan et al., 2018), EMTL can decouple the source density estimation process from the adaption phase and thus it can be used in situations where the source data is not available at the adaptation phase due to security or privacy reasons.

## 2 RELATED WORK

Zhuang et al. (2020), Kouw & Loog (2019) and Pan & Yang (2009) categorize DA approaches into instance-based and feature-based techniques. Instance-based approaches reweight labeled source samples according to the ratio of between the source and the target densities. Importance weighting methods reweight source samples to reduce the divergence between the source and target densities (Huang et al., 2007; Gretton et al., 2007; Sugiyama et al., 2007). In contrast, class importance weighting methods reweight source samples to make the source and target label distribution the same (Azizzadenesheli et al., 2019; Lipton et al., 2018; Zhang et al., 2013). Feature-based approaches learn a new representation for the source and the target by minimizing the divergence between the source and target distributions. Subspace mapping methods assume that there is a common subspace between the source and target (Fernando et al., 2013; Gong et al., 2012). Courty et al. (2017) proposed to use optimal transport to constrain the learning process of the transformation function. Other methods aim at learning a representation which is domain-invariant among domains (Gong et al., 2016; Pan et al., 2010).

Besides these shallow models, deep learning has also been widely applied in domain adaptation (Tzeng et al., 2017; Ganin et al., 2016; Long et al., 2015). DANN (Ganin et al., 2016) learns a representation using a neural network which is discriminative for the source task while cannot distinguish the source and target domains from each other. Kingma et al. (2014) and Belhaj et al. (2018) proposed a variational inference based semi-supervised learning approach by regarding the missing label as latent variable and then performing posterior inference.

## 3    NOTATION AND PROBLEM DEFINITION

We consider the unsupervised domain adaptation problem in a binary classification setting (the setup is trivial to extend to multi-class classification). Let $p(\mathbf{x}, y)$ be a joint density function defined on $\mathcal{X} \times \mathcal{Y}$, where $\mathbf{x} \in \mathbb{R}^p$ is the feature vector and $y \in \{0, 1\}$ is the label. We denote the conditional probability $p(y = 1 | \mathbf{x})$ by $q(\mathbf{x})$. A hypothesis or model is a function $h : \mathcal{X} \mapsto [0, 1]$. We define the error of $h$ as the expected disagreement between $h(\mathbf{x})$ and $q(\mathbf{x})$, i.e.,

$$\epsilon(h) = \mathbb{E}_{\mathbf{x} \sim p} |h(\mathbf{x}) - q(\mathbf{x})|. \tag{1}$$

We use superscripts s and t to distinguish the source and target domains, that is, $p^{\mathrm{s}}(\mathbf{x}, y)$ and $p^{\mathrm{t}}(\mathbf{x}, y)$ are the joint density functions in the source and target domains respectively. In general, we assume that $p^{\mathrm{s}}(\mathbf{x}, y) \neq p^{\mathrm{t}}(\mathbf{x}, y)$.

Let $\mathcal{D}^{\mathrm{s}} = \{(\mathbf{x}_i^{\mathrm{s}}, y_i^{\mathrm{s}})\}_{i=1}^{n^{\mathrm{s}}}$ and $\mathcal{U}^{\mathrm{t}} = \{\mathbf{x}_i^{\mathrm{t}}\}_{i=1}^{n^{\mathrm{t}}}$ be *i.i.d.* data sets generated from the source distribution $p^{\mathrm{s}}(\mathbf{x}, y)$ and the marginal target distribution $p^{\mathrm{t}}(\mathbf{x})$, respectively, where $n^{\mathrm{s}}$ and $n^{\mathrm{t}}$ are source and target sample sizes. The objective of unsupervised domain adaptation is to learn a model $\hat{h}$ by using labeled $\mathcal{D}^{\mathrm{s}}$ and unlabeled $\mathcal{U}^{\mathrm{t}}$, which achieves lowest error in target domain.

## 4    GENERATIVE APPROACH

Ben-David et al. (2010) proved that the error of a hypothesis $h$ in the target domain $\epsilon^{\mathrm{t}}(h)$ can be bounded by the sum of error in source domain $\epsilon^{\mathrm{s}}(h)$, the distribution distance between the two domains, and the expected $L^1$ distance between two conditional probabilities.

**Theorem 1 (Ben-David et al. (2010), Theorem 1)**  *For a hypothesis h,*

$$\epsilon^{\mathrm{t}}(h) \leq \epsilon^{\mathrm{s}}(h) + d_1(p^{\mathrm{s}}(\mathbf{x}), p^{\mathrm{t}}(\mathbf{x})) + \min\{\mathbb{E}_{\mathbf{x} \sim p^{\mathrm{s}}}|q^s(\mathbf{x}) - q^t(\mathbf{x})|, \mathbb{E}_{\mathbf{x} \sim p^{\mathrm{t}}}|q^s(\mathbf{x}) - q^t(\mathbf{x})|\}, \tag{2}$$

*where $d_1(p^{\mathrm{s}}(\mathbf{x}), p^{\mathrm{t}}(\mathbf{x})) = 2 \sup_{B \in \mathcal{B}} |\mathrm{Pr}^s(B) - \mathrm{Pr}^t(B)|$ is the twice the total variation distance of two domain distributions and $q^s(\mathbf{x})$ and $q^t(\mathbf{x})$ are the source and target probabilities of $y = 1 | \mathbf{x}$, respectively.*

In the covariate shift setting, we assume that the conditional probability $p(y | \mathbf{x})$ is invariant between the source and the target domains. Thus in the right hand side of Eq. (2), the third component will be zero, which means that the target error is bounded by the source error plus the distance between two domains. Many current unsupervised domain adaptation solutions work on how to reduce the distance between the two domain densities. Importance-sampling-based approaches manage to re-sample the source data to mimic the target data distribution, and feature-mapping-based approaches do that by learning a transformation function $\phi(\mathbf{x})$ for the source data. However, both approaches need to access source and target data simultaneously.

In this paper, we propose a domain adaptation approach based on generative models. First, we learn all multivariate densities using RNADE (Uria et al., 2013), an autoregressive version of Bishop (1994)'s mixture density nets. We found RNADE excellent in learning medium-dimensional densities, and in a certain sense it is RNADE that made our approach feasible. Second, we introduce a mediator joint density function $p^{\mathrm{m}}(\mathbf{x}, y)$ that bridges the gap between $p^{\mathrm{s}}(\mathbf{x}, y)$ and $p^{\mathrm{t}}(\mathbf{x}, y)$. Since the source distribution information is stored in the learned generative model after training, we do not need to access source data in the adaptation phase.

### 4.1    DENSITY FUNCTION

Due to recent developments in neural density estimation, we can estimate moderate-dimensional densities efficiently. In this paper, we use real-valued autoregressive density estimator (RNADE) of Uria et al. (2013). RNADE is an autoregressive version of mixture density nets of Bishop (1994) which fights the curse of dimensionality by estimating conditional densities, and provides explicit likelihood by using mixtures of Gaussians.

To estimate $p(\mathbf{x})$, let $\mathbf{x} = [x_1, x_2, \cdots, x_p]$ be a $p$ dimensional random vector. RNADE decomposes the joint density function using the chain rule and models each $p(x_i | \mathbf{x}_{<i})$ with a mixture of

Gaussians whose parameters depend on observed $\mathbf{x}_{<i}$. Formally,

$$p(\mathbf{x}) = \prod_{i=1}^{p} p(x_i | \mathbf{x}_{<i}) = \prod_{i=1}^{p} \left( \sum_{j=1}^{d} \alpha_j(\mathbf{x}_{<i}) \mathcal{N}(x_i; \mu_j(\mathbf{x}_{<i}), \sigma_j^2(\mathbf{x}_{<i})) \right), \qquad (3)$$

where $\mathbf{x}_{<i} = [x_1, \cdots, x_{i-1}]$ and $d$ is the number of Gaussian components. The weights $\alpha_j$, means $\mu_j$, and variances $\sigma_j$ are modeled by a single neural net whose architecture makes sure that the parameter $\cdot_j(\mathbf{x}_{<i})$ depends only on $\mathbf{x}_{<i}$. The neural net is trained to maximize the likelihood of the training data. We denote the RNADE model by the function $f(\mathbf{x}; \omega)$, where $\omega$ represents all the parameters (neural net weights) in RNADE, and use it to approximate $p(\mathbf{x})$. The conditional density $p(\mathbf{x}|y)$ can be estimated in the same way by just selecting $\mathbf{x}|y$ as the training data. In following sections, we denote the maximum likelihood parameters of $p^{\mathrm{s}}(\mathbf{x}|y = 0)$, $p^{\mathrm{s}}(\mathbf{x}|y = 1)$, and $p^{\mathrm{t}}(\mathbf{x})$ by $\omega_{\mathrm{s}0}$, $\omega_{\mathrm{s}1}$, and $\omega_{\mathrm{t}}$, respectively. We further denote the proportion of class 0 in the source domain by $\tau_{\mathrm{s}0} = \frac{\#\{y^{\mathrm{s}}=0\}}{n^{\mathrm{s}}}$. The full parameter vector $[\omega_{\mathrm{s}0}, \omega_{\mathrm{s}1}, \tau_{\mathrm{s}0}]$ of $p^{\mathrm{s}}(\mathbf{x}, y)$ and $p^{\mathrm{s}}(\mathbf{x})$ is denoted by $\theta_{\mathrm{s}}$.

## 4.2 THE MEDIATOR DISTRIBUTION

By Eq. (2), the target error can be bounded by the source error plus the distance between the two marginal distributions plus the expected difference in $p(y = 1|\mathbf{x})$ between two domains. This motivated us to construct a mediator distribution $p^{\mathrm{m}}(\mathbf{x}, y)$ (Figure 1) which has two properties:

- it has the same conditional distribution as the source: $p^{\mathrm{m}}(y|\mathbf{x}) = p^{\mathrm{s}}(y|\mathbf{x})$, and
- it has the same marginal distribution as the target: $p^{\mathrm{m}}(\mathbf{x}) = p^{\mathrm{t}}(\mathbf{x})$.

$$p^{\mathrm{s}} \underbrace{(\mathbf{x}, y)}_{p^{\mathrm{s}}(y|\mathbf{x})=p^{\mathrm{m}}(y|\mathbf{x})} \longrightarrow p^{\mathrm{m}} \overbrace{(\mathbf{x}, y)}^{p^{\mathrm{m}}(\mathbf{x})=p^{\mathrm{t}}(\mathbf{x})} \longrightarrow p^{\mathrm{t}}(\mathbf{x}, y)$$

Figure 1: The mediator has the same conditional probability as the source and the same marginal probability as target. According to Theorem 1, we will have $\epsilon^{\mathrm{t}}(h) \leq \epsilon^{\mathrm{m}}(h)$ for any hypotheses $h$ since the last two terms are zero.

In the covariate shift setting, we can then solve the unsupervised domain adaptation problem perfectly: i) the first property forces $p(y|\mathbf{x})$ to be the same in source and mediator distributions, and in the covariate shift setting we have $p^{\mathrm{s}}(y|\mathbf{x}) = p^{\mathrm{t}}(y|\mathbf{x})$, then this property makes $p^{\mathrm{m}}(y|\mathbf{x}) = p^{\mathrm{t}}(y|\mathbf{x})$; ii) the second property makes the marginal distributions of the mediator and the target the same, which leads to $d_1(p^{\mathrm{m}}(\mathbf{x}), p^{\mathrm{t}}(\mathbf{x})) = 0$. Under these two conditions, for any model $h$, we will have $\epsilon^{\mathrm{t}}(h) \leq \epsilon^{\mathrm{m}}(h)$ since the last two terms of Eq. (2) will be zero. Furthermore, given the mediator distribution $p^{\mathrm{m}}(\mathbf{x}, y)$, it is easy to learn the best model (Bayes classifier)

$$\hat{h}(\mathbf{x}) = \frac{p^{\mathrm{m}}(\mathbf{x}|y = 1) \, p^{\mathrm{m}}(y = 1)}{p^{\mathrm{m}}(\mathbf{x})}, \qquad (4)$$

which achieves the tightest bound for the target error. In summary, by introducing the mediator distribution $p^{\mathrm{m}}(\mathbf{x}, y)$, we can bound the target error by the mediator error. In the following sections, we will introduce how to learn $p^{\mathrm{m}}(\mathbf{x}, y)$ from $p^{\mathrm{s}}(\mathbf{x}, y)$ and $p^{\mathrm{t}}(\mathbf{x})$ using the expectation-maximization (EM) algorithm combined with a marginal constraint term.

## 5 EMTL

If we regard the missing label $y$ as a latent variable that generates observed $\mathbf{x}$ in the target domain, we can use the EM algorithm to infer $y$. We consider that the target density $p(\mathbf{x}; \theta)$ is a mixture with two components $p(\mathbf{x}|y = i; \theta)$ where $i \in \{0, 1\}$. When $\theta$ converges to its limit $\theta^*$ in EM, we can recover the joint density function $p(\mathbf{x}, y; \theta^*)$. We denote this joint density function by $p^{\mathrm{m}}(\mathbf{x}, y)$. However, this $p^{\mathrm{m}}(\mathbf{x}, y)$ may be far away from the ground truth $p^{\mathrm{t}}(\mathbf{x}, y)$. The mismatch comes from two facts: i) EM can easily converge to a bad local minimum because of a bad initialization, and ii) EM tends to find inner structure (e.g., clusters) of the data but this structure may be irrelevant

to the true label. The local minimum problem is due to parameter initialization, and the structure-label mismatching problem comes from not having a-priori information of the label. When we have a fully known source distribution $p^s(\mathbf{x}, y)$, these two issues can be solved by selecting a proper initialization plus a constraint on marginal distribution.

The first observation is that in a lot of cases we can directly use the source model in the target domain and it is better than random guess. We use this intuition to make the source model $p^s(\mathbf{x}, y)$ as the initial guess of $p^m(\mathbf{x}, y)$. Following section 4.1, we use RNADE to model $p^m(\mathbf{x} \,|y)$ and denote parameters of $p^m(\mathbf{x}, y)$ by $\theta_m = [\omega_{m0}, \omega_{m1}, \tau_{m0}]$. Initializing $p^m(\mathbf{x}, y)$ by using $p^s(\mathbf{x}, y)$ means we set $\theta_m^{(0)}$, the initial state of $\theta_m$ in the EM algorithm, to $\theta_s$. The next EM iterations can be seen as a way to fine-tune $\theta_m$ using the target data. In the next sections we will formally analyze this intuitive algorithm.

## 5.1 Analysis $\theta_m^{(1)}$

First we link the EM algorithm with initial $\theta_m^{(0)} = \theta_s$ to Theorem 1. In each iteration, EM alternates between two steps: E step defines a Q function as $Q(\theta|\theta^{(t)}) = \mathbb{E}_{y| \mathbf{x}, \theta^{(t)}} \log p(\theta; \mathbf{x}, y)$ and M step do the maximization $\theta^{(t+1)} = \arg\max_\theta Q(\theta|\theta^{(t)})$. After the first EM iteration, we have

$$\theta_m^{(1)} = \arg\max_\theta Q(\theta\,|\,\theta_m^{(0)}) = \arg\max_\theta \frac{1}{n^t} \sum_{i=1}^{n^t} \mathbb{E}_{y_i|\,\mathbf{x}_i^t, \theta_s} \log p(\mathbf{x}_i^t, y_i; \theta). \tag{5}$$

Suppose $\theta_s$ is learned perfectly from source data, which means that we can replace $p(\mathbf{x}, y; \theta_m^{(0)})$ by $p^s(\mathbf{x}, y)$. Thus the expectation operation in Eq. (5) can be written as

$$\mathbb{E}_{y_i|\,\mathbf{x}_i^t, \theta_s}[\xi] = \sum_{j\in\{0,1\}} p(y_i = j|\,\mathbf{x}_i^t; \theta_s)\xi = \sum_{j\in\{0,1\}} p^s(y_i = j|\,\mathbf{x}_i^t)\xi \tag{6}$$

for any random variable $\xi$. This expectation links the source distribution with the target. We rewrite the full expectation expression of Eq. (5) as

$$\mathbb{E}_{y_i|\,\mathbf{x}_i^t, \theta_s} \log p(\mathbf{x}_i^t, y_i; \theta) = \sum_{j\in\{0,1\}} p^s(y_i = j|\,\mathbf{x}_i^t) \log p(\mathbf{x}_i^t, y_i = j; \theta)$$
$$= -\,\mathrm{D}_{\mathrm{KL}}(p^s(y_i|\,\mathbf{x}_i^t)\|p(y_i|\,\mathbf{x}_i^t; \theta)) + \log p(\mathbf{x}_i^t; \theta) - H_{p^s}(y_i|\,\mathbf{x}_i^t), \tag{7}$$

where $H_{p^s}(y_i|\,\mathbf{x}_i^t)$ is the conditional entropy on probability $p^s$. This equation shows that the expected log-likelihood can be decomposed into the sum of three items. the first item is the negative KL-divergence between the two conditional distributions $p^s(y_i|\,\mathbf{x}_i^t)$ and $p(y_i|\,\mathbf{x}_i^t; \theta)$; the second item is the target log-likelihood $\log p(\mathbf{x}_i^t\,|\theta)$; the last item is the negative entropy of the source conditional distribution, which is irrelevant to parameter $\theta$ so can be ignored during the optimization.

Therefore, by setting $\theta_m^{(0)}$ as $\theta_s$ and maximizing the $Q$ function in the first EM iteration, we will get a $p^m(\mathbf{x}, y)$ which minimizes the KL-divergence between $p^m(y|\,\mathbf{x})$ with $p^s(y|\,\mathbf{x})$ and maximizes $\log p^m(\mathbf{x})$. Minimizing the KL-divergence reduces the third term of Eq. (2) and maximizing the log-likelihood forces $p^m(\mathbf{x})$ to move towards $p^t(\mathbf{x})$ implicitly, which reduces the second item of Eq. (2). This suggests that the Bayes classifier $p^m(y|\,\mathbf{x})$ can be a proper classifier for target domain.

## 5.2 Marginal Constraint

In the previous section, we implicitly reduce the distance between $p^m(\mathbf{x})$ and $p^t(\mathbf{x})$ by maximizing the log-likelihood of $p(\mathbf{x}; \theta)$ on the target data. To further control the target error bound Eq. (2), we explicitly add a marginal constraint for $p^m(\mathbf{x}, y)$ by minimizing the distance between the two marginal distributions. Rather than calculating $d_1(p^m(\mathbf{x}), p^t(\mathbf{x}))$ directly, we use the KL-divergence to measure the distance between two distributions since we can explicitly calculate the $p^m(\mathbf{x}_i^t)$ and $p^t(\mathbf{x}_i^t)$ by using our density estimators. Furthermore, according to Pinsker's inequality (Tsybakov, 2008), we have

$$d_1(p^m(\mathbf{x}), p^t(\mathbf{x})) \leq \sqrt{2\,\mathrm{D}_{\mathrm{KL}}(p^m(\mathbf{x})\|\,p^t(\mathbf{x}))}, \tag{8}$$

thus minimizing the KL-divergence also controls $d_1(p^{\mathrm{m}}(\mathbf{x}), p^{\mathrm{t}}(\mathbf{x}))$. Since we only have samples $\mathbf{x}_i^{\mathrm{t}}$ from the target domain, we use an *empirical* version of the KL-divergence. The marginal constraint is defined as

$$M(\theta) = \sqrt{2} \times \left( \sum_{i=1}^{n^{\mathrm{t}}} \dot{p}^{\mathrm{t}}(\mathbf{x}_i^{\mathrm{t}}) \log \frac{\dot{p}^{\mathrm{t}}(\mathbf{x}_i^{\mathrm{t}})}{\dot{p}^{\mathrm{m}}(\mathbf{x}_i^{\mathrm{t}})} \right)^{\frac{1}{2}} = \sqrt{2} \times \left( \sum_{i=1}^{n^{\mathrm{t}}} \dot{f}(\mathbf{x}_i^{\mathrm{t}}; \omega_{\mathrm{t}}) \log \frac{\dot{f}(\mathbf{x}_i^{\mathrm{t}}; \omega_{\mathrm{t}})}{\dot{p}(\mathbf{x}_i^{\mathrm{t}}; \theta)} \right)^{\frac{1}{2}}, \quad (9)$$

where $\dot{p} = p / \sum p$ and $\dot{f} = f / \sum f$ are normalized discrete distributions on the target samples.

### 5.3 Objective function of EMTL

By putting the $Q$ and $M$ functions together, we get the objective function

$$\theta^* = \arg\min_{\theta} -Q(\theta | \theta_{\mathrm{m}}^{(0)}) + \eta M(\theta) \quad (10)$$

of our generative domain adaptation approach, where $\theta_{\mathrm{m}}^{(0)} = \theta_{\mathrm{s}}$ and $\eta$ is a non-negative hyperparameter that controls the trade-off of the two terms.

In real-life scenarios, both $p(\mathbf{x})$ and $p(y|\mathbf{x})$ can be different in the source and target domains so the covariate shift assumption may be violated. To go beyond this assumption, we need to relax the constraint on $p^{\mathrm{s}}(y|\mathbf{x}) = p^{\mathrm{t}}(y|\mathbf{x})$ which is used in justifying $Q(\theta | \theta^{(0)})$. As we will show in Section 6, by setting a large $\eta$ and doing more iterations, EMTL will reduce the weight on the $Q$ function and allow us to escape from covariate shift constraints. We summarize the process of EMTL in Algorithm 1.

---

**Algorithm 1:** EMTL Algorithm

---

**Result:** EMTL classifier $p^{\mathrm{m}}(y = 1 | \mathbf{x})$

Initialize $\theta_s = [\omega_{\mathrm{s}0}, \omega_{\mathrm{s}1}, \tau_{\mathrm{s}0}]$ and $\omega_{\mathrm{t}}$ using $\mathcal{D}^{\mathrm{s}}$ and $\mathcal{U}^{\mathrm{t}}$, respectively;

Initialize $\theta_{\mathrm{m}}^{(0)}$ by $\theta_s$ and $t = 1$;

**while** $t \leq n\_itr$ **do**

$\quad | \quad \theta_{\mathrm{m}}^{(t)} = \arg\min_{\theta} -Q(\theta | \theta_{\mathrm{m}}^{(t-1)}) + \eta M(\theta)$;

$\quad | \quad t = t + 1$;

**end**

$p^{\mathrm{m}}(\mathbf{x}, y) = p(\mathbf{x}, y; \theta_{\mathrm{m}}^{(t)})$;

$p^{\mathrm{m}}(y = 1 | \mathbf{x}) = \frac{p^{\mathrm{m}}(\mathbf{x}|y=1)\, p^{\mathrm{m}}(y=1)}{p^{\mathrm{m}}(x)} = \frac{(1-\tau_{\mathrm{m}0}^{(t)}) f(\mathbf{x}; \omega_{\mathrm{m}1}^{(t)})}{(1-\tau_{\mathrm{m}0}^{(t)}) f(\mathbf{x}; \omega_{\mathrm{m}1}^{(t)}) + \tau_{\mathrm{m}0}^{(t)} f(\mathbf{x}; \omega_{\mathrm{m}0}^{(t)})}$;

---

## 6 Experiments

In this section, we present experiments on both synthetic (Section 6.1) and real-life data (Section 6.2) to validate the effectiveness of EMTL.

### 6.1 Experiments on Synthetic Data set

We study the performance of EMTL under conditional shift where $p^{\mathrm{s}}(\mathbf{x}|y) \neq p^{\mathrm{t}}(\mathbf{x}|y)$ using a variant of inter-twinning moons example (Ganin et al., 2016). In the source domain we generate an upper moon (class 0) and a lower moon (class 1) with 1000 points in each class. In the target domain, we first generate 2000 samples as in the source then rotate the data by $40°$ to make the target distribution of $\mathbf{x}|y$ different from the source. Figure 2 (left) shows the source and target distributions. In this experiments, we set the number of Gaussian components to 10 and the hidden layer dimension to 30 in the RNADE model.

We set $\eta$ to 1 and 200 to illustrate how a large $\eta$ helps the model to escape from covariate shift constraint. Figure 2 (upper right) shows the prediction results in the target data using $\eta = 1$. When $n\_itr = 0$, the EMTL classifier is the source Bayes classifier. In the upper moon, the model misclassifies the middle and the tail parts as class 1. This is because according to the source distribution, these areas are closer to class 1. The same misclassification occurs in lower moon. As $n\_itr$

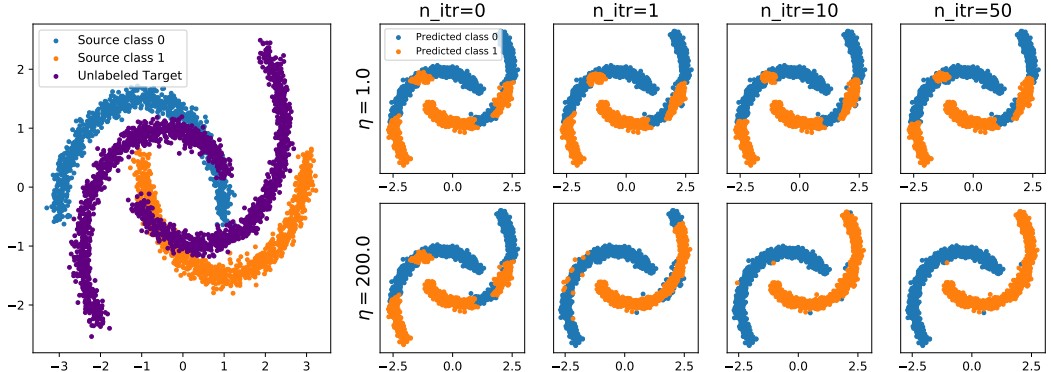

Figure 2: Inter-twining moons example. (Left) Samples from the source and target distributions where there is a $40°$ rotation in target; (Right) EMTL result on the target test data under different iterations and $\eta$s. Small $\eta$ results in a local optima. Larger $\eta$ allows the objective function to escape from the $p^{\mathrm{s}}(y|\mathbf{x}) = p^{\mathrm{t}}(y|\mathbf{x})$ constraint which is wrong in this case.

increases, the misclassification reduces slightly, because the objective function focuses more on optimizing the $Q$ function thus keeping $p(y|\mathbf{x})$ stable in each iteration. As a contrast, in Figure 2 (bottom right), when setting $\eta$ to 200, the first iteration reduces the misclassification significantly and finally the error converges to zero. By setting a large $\eta$, the conclusion of this example is two-fold: i) the $p^{\mathrm{s}}(y|\mathbf{x}) = p^{\mathrm{t}}(y|\mathbf{x})$ constraint will be relieved thus resulting in a better adaptation result, and ii) one-step iteration will increase the performance significantly thus suggesting that we do not need too many iterations. According to ii), in our following experiments the $n\_itr$ is fixed as 1. We show more experimental results using different $\eta$s in Appendix A.1 and Figure 3.

## 6.2 EXPERIMENTS ON REAL-LIFE DATA SETS

In this section, we validate EMTL on real-life data sets by comparing its performance with two standard supervised learning and three domain adaptation algorithms. The validation is conducted on three UCI data sets and the Amazon reviews data set. First, we create two benchmarks: the source RF/SVM is the model trained only using source data (as a baseline) and the target RF/SVM is the model trained only using labeled target data (as an upper bound). A random forest (RF) classifier is used on the UCI data sets and a support vector machine (SVM) is used on the Amazon reviews data set. The three DA algorithms are kernel mean matching (KMM, Huang et al. (2007)), subspace alignment (SA, Fernando et al. (2013)) and domain adversarial neural network (DANN, Ganin et al. (2016)). For the UCI data sets, both KMM and SA are based on RF and for Amazon reviews data set SVM is used. In KMM, we us an RBF kernel with the kernel width set as the median distance among the data. In DANN, $\lambda$ is fixed as 0.1. In EMTL, we set the number of components to 5 and the hidden layer size to 10 for RNADE model and $\eta$ to 1. For each transfer task, five-fold cross validation (CV) is conducted. In each CV fold, we randomly select 90% source samples and 90% target samples respectively to train the model. We average the output of the five models and calculate the 95% confidence interval of the mean. For the UCI tasks, ROC AUC score is the used metric since we are dealing with imbalanced classification tasks. For Amazon reviews tasks accuracy is the used metric. Table 1 and 2 summarize the experimental results. Numbers marked in bold indicate the top performing DA algorithms (more than one bold means they are not significantly different).

**UCI data sets.** Three UCI data sets (Abalone, Adult, and Bank Marketing) are used in our experiments (Dua & Graff, 2017; Moro et al., 2014). We preprocess the data first: i) only select numerical features; ii) add uniform noise to smooth the data from integer to real for Adult and Bank data sets. Since the original goal in these data sets is not transfer learning, we use a variant biased sampling approach proposed by Gretton et al. (2009) and Bifet & Gavaldà (2009) to create different domains for each data set. More precisely, for each data set we train a RF classifier to find the most important feature, then sort the data along this feature and split the data in the middle. We regard the first 50% (denoted by A) and second 50% (denoted by B) as the two domains. When doing domain

Table 1: Experimental results on UCI data sets. AUC(%) is used as a metric.

| Task | Source RF | Target RF | KMM | SA | DANN | EMTL |
|---|---|---|---|---|---|---|
| Abalone A → B | $67.1 \pm 1.1$ | $72.7 \pm 0.5$ | $66.5 \pm 2.2$ | $67.8 \pm 0.6$ | $67.5 \pm 0.4$ | $65.7 \pm 2.8$ |
| Abalone B → A | $67.5 \pm 1.2$ | $81.2 \pm 0.4$ | $59.4 \pm 4.6$ | $68.5 \pm 2.1$ | $69.5 \pm 0.7$ | $70.8 \pm 0.7$ |
| Adult A → B | $84.4 \pm 0.2$ | $84.8 \pm 0.2$ | $83.4 \pm 0.4$ | $82.8 \pm 0.2$ | $84.7 \pm 0.1$ | $84.8 \pm 0.3$ |
| Adult B → A | $82.1 \pm 0.1$ | $83.1 \pm 0.1$ | $81.3 \pm 0.4$ | $81.0 \pm 0.2$ | $82.8 \pm 0.3$ | $82.7 \pm 0.4$ |
| Bank A → B | $70.1 \pm 0.3$ | $81.5 \pm 0.1$ | $69.3 \pm 1.1$ | $70.4 \pm 0.9$ | $70.8 \pm 0.5$ | $70.5 \pm 1.7$ |
| Bank B → A | $76.7 \pm 0.7$ | $83.0 \pm 0.6$ | $74.8 \pm 0.5$ | $76.6 \pm 0.4$ | $78.4 \pm 0.2$ | $79.3 \pm 0.8$ |

Table 2: Experimental result on Amazon reviews data set. Accuracy(%) is used as a metric.

| Task | Source SVM | Target SVM | KMM | SA | DANN | EMTL |
|---|---|---|---|---|---|---|
| B → D | $80.0 \pm 0.0$ | $79.9 \pm 0.1$ | $79.7 \pm 0.2$ | $79.9 \pm 0.1$ | $79.9 \pm 0.0$ | $79.5 \pm 0.1$ |
| B → E | $70.3 \pm 0.1$ | $72.4 \pm 0.2$ | $72.9 \pm 0.2$ | $73.0 \pm 0.2$ | $69.7 \pm 0.3$ | $71.5 \pm 0.2$ |
| B → K | $75.7 \pm 0.1$ | $76.2 \pm 0.1$ | $76.3 \pm 0.0$ | $76.1 \pm 0.1$ | $75.7 \pm 0.1$ | $76.0 \pm 0.1$ |
| D → B | $75.5 \pm 0.0$ | $75.5 \pm 0.1$ | $75.3 \pm 0.1$ | $75.3 \pm 0.1$ | $75.4 \pm 0.1$ | $75.7 \pm 0.0$ |
| D → E | $71.8 \pm 0.1$ | $74.2 \pm 0.1$ | $74.6 \pm 0.1$ | $74.4 \pm 0.0$ | $71.5 \pm 0.1$ | $72.3 \pm 0.2$ |
| D → K | $75.7 \pm 0.1$ | $77.0 \pm 0.0$ | $76.8 \pm 0.1$ | $77.4 \pm 0.1$ | $75.6 \pm 0.3$ | $76.1 \pm 0.2$ |
| E → B | $70.3 \pm 0.1$ | $71.0 \pm 0.1$ | $71.8 \pm 0.1$ | $71.4 \pm 0.1$ | $70.5 \pm 0.0$ | $69.5 \pm 0.3$ |
| E → D | $72.2 \pm 0.0$ | $73.1 \pm 0.1$ | $73.1 \pm 0.3$ | $73.1 \pm 0.1$ | $72.1 \pm 0.1$ | $72.7 \pm 0.2$ |
| E → K | $85.8 \pm 0.1$ | $86.2 \pm 0.0$ | $83.6 \pm 0.8$ | $86.0 \pm 0.1$ | $85.8 \pm 0.2$ | $85.3 \pm 0.1$ |
| K → B | $71.5 \pm 0.0$ | $71.6 \pm 0.1$ | $71.4 \pm 0.2$ | $71.5 \pm 0.0$ | $71.3 \pm 0.1$ | $71.6 \pm 0.1$ |
| K → D | $70.6 \pm 0.0$ | $71.7 \pm 0.2$ | $72.6 \pm 0.3$ | $72.4 \pm 0.1$ | $70.6 \pm 0.1$ | $71.6 \pm 0.2$ |
| K → E | $83.9 \pm 0.0$ | $84.3 \pm 0.0$ | $84.2 \pm 0.1$ | $84.3 \pm 0.1$ | $84.0 \pm 0.1$ | $83.9 \pm 0.2$ |

adaptation, we use 75% of the target domain samples to train the model and use the other 25% target domain samples as test data. Finally, we use normal quantile transformation to normalize the source and target data sets respectively. Table 3 Appendix A.2 summarizes the features of the data sets we created for the experiments. Table 1 shows the results on the test data for UCI data sets. We find that the performance of EMTL is not significantly different from DANN in all tasks (remember that our goal was not the beat the state of the art but to match it, without accessing the source data at the adaptation phase). On the two Adult tasks and Bank B → A, although the average score of EMTL is less than that of Target RF, the differences are small.

**Amazon reviews.**   This data set (Ganin et al., 2016) includes four products, books (B), DVD (D), electronics (E) and kitchen (K) reviews from the Amazon website. Each product (or domain) has 2000 labeled reviews and about 4000 unlabeled reviews. Each review is encoded by a 5000-dimensional feature vector and a binary label (if it is labeled): 0 if its ranking is lower than three stars, and 1 otherwise. We create twelve transfer learning tasks using these four domains. As RNADE is not designed for ultra high dimensional cases, we overcome this constraint by reducing the number of features from 5000 to 5 using a feed forward Neuronal Network (FNN). More precisely, for each task we train a 2-hidden layer FNN on the source data. Then, we cut the last layer and we use the trained network to encode both source and target to 5 dimensions. Table 2 shows the results on the test data for Amazon reviews data set. We notice that EMTL is slightly better than DANN in most of the tasks and still comparable with both KMM and SA.

## 7    CONCLUSIONS AND FUTURE WORK

In this paper, we have presented a density-estimation-based unsupervised domain adaptation approach EMTL. Thanks to the excellent performance of autoregressive mixture density models (e.g., RNADE) on medium-dimensional problems, EMTL is competitive to state-of-the-art solutions. The advantage of EMTL is to decouple the source density estimation phase from the model adaptation phase: we do not need to access the source data when adapting the model to the target domain. This property allows our solution to be deployed in applications where the source data is not available after preprocessing. In our future work, we aim to extend EMTL to more general cases, including high-dimensional as well as more complex data (e.g., time series).

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

## A  APPENDIX

### A.1  INTER-TWINNING MOONS EXAMPLE

We test three $\eta$ settings and compare the corresponded AUC and accuracy in Appendix Figure 3. We find that as $n\_itr$ increase, the AUC and accuracy will increase too. In each fixed $n\_itr$, a larger $\eta$ always has higher AUC and accuracy.

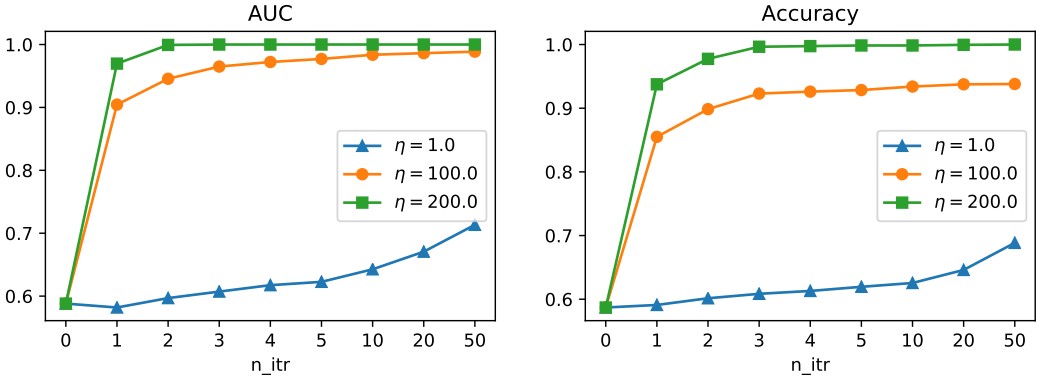

Figure 3: In inter-twinning moons example, as $\eta$ increase, both AUC and accuracy will increase.

## A.2 UCI EXPERIMENTS

We summarize the size and class ratio information of UCI data sets in Appendix Table 3.

Table 3: UCI data sets

| Task | Source | Target | Test | Dimension | Class 0/1(Source) |
|---|---|---|---|---|---|
| Abalone A → B | 2,088 | 1,566 | 523 | 7 | 24% vs. 76% |
| Abalone B → A | 2,089 | 1,566 | 522 | 7 | 76% vs. 24% |
| Adult A → B | 16,279 | 12,211 | 4,071 | 6 | 75% vs. 25% |
| Adult B → A | 16,282 | 12,209 | 4,070 | 6 | 77% vs. 23% |
| Bank A → B | 22,537 | 17,005 | 5,669 | 7 | 97% vs. 03% |
| Bank B → A | 22,674 | 16,902 | 5,635 | 7 | 80% vs. 20% |

**Parameter settings in UCI data sets.**  We enumerate the parameter settings on UCI experiment here.

- Random forest models with 100 trees are used as the classifier.

- For DANN, we set the feature extractor, the label predictor, and the domain classifier as two-layer neural networks with hidden layer dimension 20. The learning rate is fixed as 0.001.

- For EMTL, we fix the learning rate as 0.1 except for the task Abalone B → A (where we set it to 0.001) as it did not converge. As mentioned in section 6.1, we only do one EM iteration.

**Parameter settings in Amazon reviews dataset.**  We enumerate the parameter settings choice of Amazon reviews experiment here.

- SVM has been chosen over RF because it showed better results in the case of Amazon reviews experimentation

- We run a grid search to find the best C parameter for SVM over one task (from books to dvd) the best result $C = 4.64E - 04$ is then used for all tasks and for source svm, target svm, KMM and SA solutions.

- For DANN, we set the feature extractor, the label predictor, and the domain classifier as one-layer neural networks with hidden layer dimension 50. The learning rate is fixed as 0.001.

- FNN is composed of 2 hidden layers of dimensions 10 and 5 (the encoding dimension). we added a Gaussian Noise, Dropout, Activity Regularization layers in order to generalize better and guarantee better encoding on target data.

- For EMTL, we fix the learning rate as 0.001 and only do one EM iteration.

Note that the presented result of Amazon reviews data set in Table 2 have been rounded to one digit. This explains why the 95% confidence interval of the mean is sometimes equal to 0.0 and why some values are not in bold.

