# OpenReview forum: "EMTL: A Generative Domain Adaptation Approach"
_ICLR.cc/2021/Conference — Reject_

### Official Review · AnonReviewer2 · 2020-10-26
**Official Blind Review2**

**Rating:** 3
**Confidence:** 5

**Review:**

In this paper, the authors propose generative domain adaptation approach called EMTL. The key idea is to model a mediator distribution which can approximate the true target joint distribution. Specifically, the authors apply an E-M strategy to infer the model parameters. Experimental studies are done on both synthetic and real-world datasets.

The paper is well-organized and easy to follow. My major concern is on the significance of the paper, which is not significantly novel especially compared with the recent DNN-based domain adaptation methods. Moreover, there are some technical flaws, which need to be further clarified. Here are the detailed comments:

(1)	One motivation of the paper is the sensitivity of the source data. Due to security or privacy issues, source data may not be accessible. While it is a practical and nice point of motivation, the paper misses one important research line on federated learning that is specially proposed for privacy issues. It is necessary to discuss with some federated learning related works.

(2)	Regarding the access of the source data, the proposed method still requires the source data to do source density estimation. In this sense, I am not convinced by the claim on the privacy preservation. From Algorithm 1, it can be clearly seen that D^s is still used for the initialization of \theta_s.

(3)	The authors highlight no access of source data in the adaptation phase. Could you elaborate on what is the adaptation phase? Based on my understanding, the whole Algorithm 1 is for adaptation, but it still needs source data. Is it better to claim that the proposed method only requires source model parameters trained previously?

(4)	The related work section can be further improved, by discussing more on both subspace-based and deep-learning based domain adaptation methods.

(5)	The key idea is based on Theorem 1, and aims to build a mediator distribution to approximate the target joint distribution. Most of existing domain adaptation methods share the same idea although they are not generative models, please highlight the main advantage of proposed generative model over existing subspace-based and deep-learning based methods.

(6)	Regarding Eq. (5), why \theta_s is used in the subscription of \mathbb{E}? It should be \thetha_m^(0), right? Moreover, how to obtain y_i for each target data point?

(7)	For Eq. (5), does it only hold for the first iteration where the source parameters are used as the initialization? For the following iterations, are you still using p^s(y_i = j | x_i^t)? or using p^m(y|x)? If the latter is used, does it mean y_i is updated in each iteration?

(8)	The proposed synthetic dataset is very naïve. It is more convincing to test on more complex datasets with higher dimensionality data.  For the real-world datasets, there are a lot of benchmark datasets for domain adaptation, e.g., office 31, office-caltech 10, and office-home etc. It is more convincing to test on these well-known datasets. More importantly, please compare with more state-of-the-art baselines, on both subspace-based and deep learning based. Even on the reported datasets, the improvements of EMTL over SA and DANN (these 2 are not state-of-the-art) are marginal.

Update: Thanks for the authors' response. After reading the response and the other reviewers' comments, I think the paper needs to be further improved, and thus I will keep my score.

---

> ### Author Response · Authors · 2020-11-18
> **Answer to reviewer #2**
>
> We thank you very much for your thoughtful review. Below are detailed explanations of the points you raised.
>
> - 1-3: We apology for having put too much focus on privacy and thank you for pointing out the logical flaws. We updated the abstract. As kindly mentioned by other reviewer, our goal is the same as the one in [1], i.e., "*tackles a practical setting where only a trained source model is available and investigates how we can effectively utilize such a model without source data to solve UDA problems.*"[1]  As a side effect of not accessing the source data during the adaptation phase, we may have put some light on privacy theory, misleading the reader on the general direction of this research. Privacy is not our prime focus in this paper. We rewrote part of the introduction in order to make this really clear.
>
> - 4: We updated the related work section, we added:
>     - Liang, Jian, et al. "Do We Really Need to Access the Source Data? Source Hypothesis Transfer for Unsupervised Domain Adaptation." ICML (2020).
>     - Yang, Shiqi, et al. "Unsupervised Domain Adaptation without Source Data by Casting a BAIT."
>     - Peng, Xingchao, et al. "Federated adversarial domain adaptation. (ICLR2020).
>     - Song, et al. Privacy-Preserving Unsupervised Domain Adaptation in Federated Setting. IEEE Access (2020).
>     -  Rui, Shu, et al. A DIRT-T Approach to Unsupervised Domain Adaptation. ICLR (2018).
>     - Kumar, et al. Understanding Self-Training for Gradual Domain Adaptation. ICML(2020)
>
> - 5:
> The advantage of generative model is in allowing to explicitly minimize the right hand side of theorem 1. This is the biggest difference between EMTL and other current solutions. Actually, most of them, e.g., DANN, SHOT and DIRT-T, are from the representation side rather than the distribution side to minimize the rhs of the theorem 1.  However, DANN needs to access both source and target data at the same time, SHOT and DIRT-T use on "clustering" assumptions to further constrain the representation learning during adaptation phase, which is a very great heuristic but not an explicitly way.
>
> - 6-7:
> We use several RNADE models in order to estimate the density of the datasets regarding of the labels: 1) $p^s(x|y=0)$ consists in one RNADE model with parameter $\omega_{s0}$ and 2) $p^s(x|y=1)$ is another RNADE model with parameter $\omega_{s1}$. As we do not have access to target dataset labels, we learn a single RNADE model to have an estimation of $p^t(x)$. The goal of our transfer learning method is to infer the true distribution $p^t(x,y)$ in order to accurately classify the target dataset. We introduce the mediator distribution $p^m(x,y)$, composed of 2 RNADE models (one for each label), as the surrogate of true $p^t(x, y)$. We summarize the parameters we used to make them clearer:
>     - $\theta_s = [\omega_{s0}, \omega_{s1}, \tau_{s0}]$ is the parameter for $p^s(x,y)$. It is learned by using source data and is fixed during the EM steps.  It includes three parts:
>         - $\omega_{s0}$ is the RNADE parameters of $p^s(x|y=0)$, we learned it by using source data $x|y=0$;
>          - $\omega_{s1}$ is the  RNADE parameters of $p^s(x|y=1)$, we learned it by using source data $x|y=1$;
>          - $\tau_{s0}$ is the ratio of class 0 in source data;
>     - $\omega_{t}$ is the RNADE parameter of $p^t(x)$, we learned it by using full unlabeled target data $x$;
>     - $\theta_m =  [\omega_{m0}, \omega_{m1}, \tau_{m0}]$ is the parameter for $p^m(x,y)$. This is unknown and is the one we want to learn. It also includes three parts (just as $\theta_s$):
>         - $\omega_{m0}$ is the RNADE parameters of $p^m(x|y=0)$;
>         - $\omega_{m1}$ is the  RNADE parameters of $p^s(x|y=1)$;
>         - $\tau_{m0}$ is the ratio of class 0;
>
>  Eq. (5) is indeed only valid for the first iteration. At $k+1$-th iteration, $\theta_m^{(k)}$ is used in order to compute $\theta_m^{(k+1)}$. The $y_i$ in Eq. (5) serves as the latent variable as regular EM did. The whole section is rewritten in a much clearer way in order to avoid confusion for the reader, all the previous points are now explicitly and clearly stated.
> - 8: We are working on more benchmark datasets as well as adding more SOTA DA algorithms and we will update the experimental results as soon as possible.
>
> ### Reference
> [1]: Do We Really Need to Access the Source Data? Source Hypothesis Transfer for Unsupervised Domain Adaptation. Jian Liang, Dapeng Hu, Jiashi Feng. ICML 2020.

---

### Official Review · AnonReviewer3 · 2020-10-27
**An interesting motivation, but more work would be needed.**

**Rating:** 5
**Confidence:** 4

**Review:**

This paper proposes a novel method for Unsupervised Domain Adaptation (UDA) when the source domain's privacy should be preserved. The authors propose EMTL, which is a generative method using multivariate densities using RNADE (Uria et al., 2013) and a mediator joint density function bridging both source and target domains. EMTL achieves comparable performances to those of DANN (Ganin et al., 2016) on a single dataset.

**Pros**

- Unique motivation for UDA and privacy-preserving.
- Well formulated method using RNADE and a mediator density function. In the adaptation phase, the source domain data can be deleted.

**Cons**

- There is a closely related paper for privacy-preserving UDA (Song et al., 2020) before the deadline of ICLR 2021. The method by Song et al. utilized a framework of federated learning and encryption. Thus the approaches are different from each other, but the motivation for privacy-preserving is close. The authors should compare them quantitatively.

Song et al. Privacy-Preserving Unsupervised Domain Adaptation in Federated Setting. IEEE Access, Vol. 8, pp.143233-143240, 2020.

- Although the adaptation phase does not require the source domain data, a probabilistic function $p^m(y|x)$ should be available. The reviewer just concerns if model inversion attacks, such as (Fredrikson et al., 2015), violate the source domain's privacy.

Fredrikson et al. Model Inversion Attacks that Exploit Confidence Information and Basic Countermeasures. In CCS, 2015.

- It is reasonable to compare ETML to DANN since both methods have conceptually similar characteristics: matching the data distribution and learning the posterior probabilities of the label given a sample. However, as the authors referred to in the main text, several methods have similar characteristics and much better performance than DANN. It is good to know if ETML is complementary to those methods through further experiments.
- Experiments on a single real dataset are difficult to convince about the generality of UDA performance. Additional experiments on other datasets such as visual ones can strengthen the generality.

**Overall rating**

The reviewer is leaning toward rejection, although the motivation is clear. The rating can be upgraded if the authors can solve the cons above.

**Additional comment after rebuttal**

Happy to hear that the authors plan to upgrade their draft. Since the submitted paper is not updated, the reviewer keeps the first rating but also looks forward to read a revised version in another conference or journal.

---

> ### Author Response · Authors · 2020-11-18
> **Answer to reviewer #3**
>
> 1. We apology for having put too much focus on privacy and thank you for pointing out the logical flaws. We updated the abstract. As kindly mentioned by other reviewer, our goal is the same as the one in [1], i.e., "tackles a practical setting where only a trained source model is available and investigates how we can effectively utilize such a model without source data to solve UDA problems." As a side effect of not accessing the source data during the adaptation phase, we may have put some light on privicy theory, misleading the reader on the general direction of this research. Privacy is not our prime focus in this paper, which remains DA. We rewrote part of the introduction in order to make this really clear.
>
> 2. We are currently performing experiments in order to better justify our approach. We will update the paper really soon with extensive results.
>
>
>
> [1] Do We Really Need to Access the Source Data? Source Hypothesis Transfer for Unsupervised Domain Adaptation. Jian Liang, Dapeng Hu, Jiashi Feng. ICML 2020.

---

### Official Review · AnonReviewer1 · 2020-10-28
**Review #1**

**Rating:** 3
**Confidence:** 5

**Review:**

--------Updates after rebuttal-----------

Since the author did not propose an updated paper and new experiments. I keep my original score.

---------------------------------------------------



Summary:
This paper proposed a generative domain adaptation (DA) approach under covariate shift. Different from previous domain discriminator methods, this paper introduced a mediator distribution and adopted an autoregressive approach (RNADE) to estimate the distribution density. Empirical results on simple datasets (UCI and Amazon) verified its practical benefits.

------------------------------------------------------
Overall review

Pros:

[1] As far as I know, this is the first paper that used the autoregressive approach in DA.

[2] The proposed adaptation algorithm does not require accessing the source data at the adaptation phase, which has some practical potential.

[3] The high-level idea seems logical and correct (But some technical details seem problematic.)

Cons:

[1] The motivation of the proposed approach is unclear: it seems a simple plug-in approach with RNADE in DA. A thorough analysis is lacking.

[2] The empirical significance of the paper is rather limited: the paper did not effectively show its practical utility.

[3]  Some technical details are difficult to follow or flawful.

Based on these reasons, I recommend rejection.

--------------------------------------------------
Detailed explanations

[1] Motivation

I am rather confused about the motivation of the proposed approach. As for the generative model, the particular reason to use RNADE is unclear. Is the discriminator unable to solve the source-target separation issue? An alternative approach is to train a model on source only and apply the unlabelled target for fine-tuning. (see recent paper [1]). Discussion on the benefits of these settings is highly expected.

[2] Experiments

Since it is an empirical paper, I am most concerned about the empirical results.

[a] The current results are rather limited. The author only evaluated on UCI and Amazon review dataset. Both are simple datasets and linear models can achieve good results.

[b] The compared baselines are NOT SOTA. DANN is the standard baseline.

[3] Technical details

[a] The RNADE is a high time complexity approach for high dimensional data. I would like to see an empirical and theoretical discussion on the high-dimensional dataset (such as the image)

[b] The notation in the EM algorithm is rather confusing and difficult to follow.
Besides, Eq (7) is the log-MLE approach, then it can be naturally decomposed in three terms. Eq (9) is not correct, $p(x)$ should be a continuous function (not discrete). Using the empirical counterpart to estimate the KL divergence is problematic in the high dimensional dataset.

[c] Sec 5.3 “As we will show in Section 6, by setting a large $\eta$ and doing more iterations, EMTL will reduce the weight on the Q function and allow us to escape from covariate shift constraints”. This discovery is really important and interesting. I think it deserves a better justification.

--------------------------------------------
Suggestions

I suggest extensive empirical results for showing the effectiveness of the proposed approach.


[1] Do We Really Need to Access the Source Data? Source Hypothesis Transfer for Unsupervised Domain Adaptation. ICML 2020

---

> ### Author Response · Authors · 2020-11-18
> **Answer to reviewer #1**
>
> 1. Discriminator cannot fully replace the density estimator in EMTL. The density function $p(x,y;\theta)$ is tunable thus allows us to optimize $D_{\text{KL}}(p^s(y|x), p(y|x;\theta))$, $\log p(x;\theta)$ and $D_{\text{KL}}(p^t(x), p(x;\theta))$ at the same time. Discriminator (let's say a neural network $f(x)$) can be used as approximator for $p(y|x)$, but that is not enough. Discriminator does not have capability to optimize $D_{\text{KL}}(p^t(x), p(x))$, thus we further need another module like MMD as a regularizer. For example, DANN includes both classifier (discriminator) and adversarial item (MMD module).  Thus the density estimator is crucial in our algorithm in order to accurately perform the computations described in section 5.1 and 5.2.  However RNADE is not the only possible choice, any density estimator may be used as an alternative. In our work we choose RNADE because it is a powerful, yet simple, density estimator in moderate-dimensional problem. We will better clarify our approach regarding density estimation in the paper.
> 2. Both SHOT and EMTL solve the same problem, i.e., the source data is unaccessible in adaptation phase. But SHOT and EMTL tackle the problem from two different directions: SHOT fixes the classifier and tunes the feature extractor in target. Whereas EMTL fixes the feature part but tunes the classifier. SHOT includes a global diverse loss in the objective function to encourage the target model to generate a diversity results. However, this assumption may be violated in a lot of real applications where the target class distributions can be imbalanced and unknown. As a contrast, EMTL doesn't have this issue.
> 3. We did choose to use a naïve empirical approximation for the KL-divergence in Eq.(9). As correctly noted, this empirical estimator is indeed not well suited in high dimension setting. Closed-form expression or better estimators exist for GMM, but cannot be straightforwardly applied to RNADE. We leave the extension to tighter estimator and/or upper bound for future work.
> 4. The hyperparameter $\eta$ leverage the trade-off between $D_{\text{KL}}(p^s(y|x), p(y|x))$ and $D_{\text{KL}}(p^s(x), p(x)) $:
>     - When setting $\eta$ as 0, optimizing objective function results to a $p(x,y)$ which has $p(y|x)=p^s(y|x)$. In covariate shift case where we assume $p^s(y|x)=p^t(y|x)$, thus the learned $p(y|x)$ is $p^t(y|x)$.
>     - When setting a big $\eta$, the first term becomes negligible and EMTL focuses more on reducing the distance between $p(x)$ and $p^t(x)$. As "$p(y|x)=p^s(y|x)$" constraint is weak thus EMTL can escape from covariate shift setting and fit to more general cases.
> In summary, EMTL allows us to balance well between these two settings. It is interesting to notice that in most of our experiments, the optimal eta is located precisely in the "middle" of these two worlds. We clarified this section and added the etas used in the experimental section to put more light on this important property of EMTL.
> 5. We are currently performing experiments in order to better justify our approach. We will update the paper really soon with extensive results.

---

> > ### Comment · AnonReviewer1 · 2020-11-19
> > **Feedback**
> >
> > Thanks for your quick response!
> > 1. Thanks for your explanation. Maybe my review is not clear, the `discriminator`  in my review is referred to domain discriminator, not the classifier. I am wondering about the difference between domain discriminator and your approach.
> >
> > 2. Now I have a better understanding of the difference between the SHOT and EMTL. Thanks!
> >
> > 3. Yes. I think the biggest concern in the proposed approach is the vanilla approach in estimating KL divergence for the high dimensional dataset.
> >
> > 4. The covariate assumption is difficult to achieve for the deep learning-based approach. e.g. considering the source domain SVHN and target domain MNIST, the underlying label generation distribution can be really different.
> >
> > 5. I am looking forward to your additional experiments.

---

### Official Review · AnonReviewer4 · 2020-10-29
**Nice idea but privacy properties, conceptual intuition, and relation to other methods, is unclear**

**Rating:** 4
**Confidence:** 3

**Review:**

Update after author response: I appreciate the clarifications, but given the lack of comparisons or discussion to related prior methods (at least Liang et al 2020 or some alternative equivalent), I cannot recommend acceptance at this point. The authors did not submit a paper revision as well. I think the idea seems promising, so don't take this as a critique of the research direction.

#########################################################################

Summary:

This paper tackles the problem of unsupervised domain adaptation, where there may be privacy constraints on the source, so we have access to a source model but not the source data. Their approach first learns a generative model p(x, y | theta_s) for the source. They use this to pseudolabel every target example p(y^t_i | x^t_i, theta_s), and then fit a new generative model to these examples. They iterate this process multiple times to get their final model p(x, y | theta), from which they can predict p(y | x, theta).

#########################################################################

Reasons for score:

I think this is a cool idea, but the paper in its current form seems incomplete. One of the goals is to preserve privacy of the source dataset, but it seems unlikely that a generative model p(x, y | theta_s) learned on the source preserves privacy. In particular, we could effectively “generate” the source and get access to private information. Is there any prior work that suggests that this is not the case? This seems to be a key point, since barring privacy concerns their method doesn’t seem to do better than baselines. The conceptual explanation for the method seems unclear and incomplete (more below). The method could also be better tied into /compared with the existing privacy preserving domain adaptation literature, and the self-training literature. Overall, this is a promising and interesting idea, and with more work would be good to publish.

#########################################################################

Pros:

- I like the idea of using a generative model for x and y, and repeatedly applying that by pseudolabeling the target and self-training.

- Their synthetic experiments show some promise, and on some of the UCI datasets they seem to do well.


#########################################################################

Cons:

- No explanation given for why a generative model p(x, y | theta_s) learned on the source preserves privacy, when having a generative model would allow us to generate samples and potentially get sensitive information. This seems to be a key point of the paper, since experimental results are not better than baselines, but the argument is that those don’t preserve privacy.

- There are many other related alternatives. One prominent alternative is self-training or pseudolabeling based approaches, which train a classifier on the source, and use it to pseudolabel the target, then training a regularized classifier on the pseudolabeled data. This, and related methods such as entropy minimization, has been used in the context of domain adaptation (Shu et al, Kumar et al), also in the context of privacy preserving domain adaptation (Liang et al). These methods are highly scalable as well, and all the above papers use multi-layer neural networks and high dimensional datasets. Would be good to compare with at least some existing baseline.

- The conceptual / mathematical explanation seems unclear and incomplete. If there is only covariate shift, then P_m and P_t should be identical. This is because P_t and P_m have the same distribution over x. P_m has the same y | x distribution as P_s, but in covariate shift this is the same as P_t. So it’s unclear what P_m is doing. Section 5.1 argues that the EM training objective enforces P_m to have similar x distribution to P_t and similar y | x distribution to P_s. However, there isn’t a proper explanation of why this objective is a good thing. There seems to be an attempt to explain this in Section 4.2, but in the case of no covariate shift, the Bayes classifier on the source = Bayes classifier on the target, so it suffices to simply train a model on the source.

#########################################################################

Questions and things to improve:

Please address questions and comments listed in cons.

#########################################################################

References mentioned:

A DIRT-T Approach to Unsupervised Domain Adaptation. Rui Shu, Hung H. Bui, Hirokazu Narui, Stefano Ermon. ICLR 2018.

Understanding Self-Training for Gradual Domain Adaptation. Ananya Kumar, Tengyu Ma, Percy Liang. ICML 2020.

Do We Really Need to Access the Source Data? Source Hypothesis Transfer for Unsupervised Domain Adaptation. Jian Liang, Dapeng Hu, Jiashi Feng. ICML 2020.

---

> ### Author Response · Authors · 2020-11-24
> **Answer to reviewer #4**
>
> - We apology for having put too much focus on privacy and thank you for pointing out the logical flaws. We updated the abstract. As kindly mentioned by other reviewer, our goal is the same as the one in [1], i.e., "tackles a practical setting where only a trained source model is available and investigates how we can effectively utilize such a model without source data to solve UDA problems."  As a side effect of not accessing the source data during the adaptation phase, we may have put some light on privacy theory, misleading the reader on the general direction of this research. Privacy is not our prime focus in this paper, which remains DA. We rewrote part of the introduction in order to make this really clear.
>
> - We thank you for the recommending these papers. Although, their relatedness to our work, we unfortunately missed them during our paper survey phase.  We  have extended our related work to include them. We are also working on adding these SOTA algorithms in our experiments and conducting a more comprehensive comparison: 1) we are adding these algorithms on UCI and amazon experiments. 2) image experiments listed in these papers are added as well. We will update the experimental results as soon as possible.
>
> - We thank you for pointing out the unclear and incomplete part in mathematical explanation. We have modified and re-arranged the paper so that $p^m$ and also the EMTL concept are much more clear.  $p^m(x,y)$ aims to minimize $E_{x \sim p^t(x)} D_{KL}(p^s(y|x), p(y|x;\theta_m))$ and maximize  $E_{x \sim p^t(x)} \log p(x;\theta_m)$ (same to minimize $D_{KL}(p^t(x),p^m(x))$). Thus, in the paper we mentioned that  $p^m(x,y)$ bridges the gap between source and target. The hyper parameter $\eta$ serves to balance  $E_{x \sim p^t(x)} D_{KL}(p^s(y|x), p(y|x;\theta_m))$ and $E_{x \sim p^t(x)} \log p(x;\theta_m)$ and actually its role is similar to $\lambda$ in DANN (i.e., the hyper parameter controlling the domain regularizer [3]). In the case where $\eta=0$, EMTL is exactly a source classifier. This will be formally proved in the updated paper. This $\eta$ parameter leverages the trade-off between those two terms and hence between two different regimes.
>
> ### Reference
> [1]: Do We Really Need to Access the Source Data? Source Hypothesis Transfer for Unsupervised Domain Adaptation. Jian Liang, Dapeng Hu, Jiashi Feng. ICML 2020.
>
> [2] Ben-David, Shai, et al. "A theory of learning from different domains." Machine learning 79.1-2 (2010): 151-175.
>
> [3] Ganin, Yaroslav, et al. "Domain-adversarial training of neural networks." The Journal of Machine Learning Research 17.1 (2016): 2096-2030.

---

### Decision · Program_Chairs · 2021-01-07
**Final Decision**

**Decision:**

Reject

**Comment:**

This work proposes an EM type of approach for domain adaptation under covariate shift. The approach well motivated and developed and experimentally evaluated on synthetic data.

Pro:
- The EM type of framework is simple and natural and  promising direction for DA, which should be explored and analyzed further.

Con:
- The presentation is highly overselling the results. Both in terms of the generality of the findings and in terms references to privacy preserving properties. Both would need a solid formal analysis which this submission does not provide.
- Several reviewers have stated that, while the authors promised updates to their manuscript during the author response phase, no such updated submission has been made.
- The work bases their approach by referring to a well known theoretical DA bound by Ben-David et al (2010). The theorem is not stated correctly. The most important component in that work is to restrict the models to a class of bounded capacity.
- The claim of the authors of "solving the problem" under covariate shift are overstated. It is reasonable to expect that the authors provide a more thorough analysis of the limitations or their approach, that is, clearly state the conditions under which it would succeed and fail. Below are some references on lower bounds of DA under covariate shift.
- Given that the theoretical analysis is limited, a more thorough experimental exploration would be expected.

Refs on difficulty of DA learning under covariate shift and bounded d_H distance:

Shai Ben-David, Ruth Urner:
On the Hardness of Domain Adaptation and the Utility of Unlabeled Target Samples. ALT 2012: 139-153

Shai Ben-David, Tyler Lu, Teresa Luu, Dávid Pál:
Impossibility Theorems for Domain Adaptation. AISTATS 2010: 129-136